# Climate Change and Diurnal Warming: Impacts on the Growth of Different Vegetation Types in the North–South Transition Zone of China

**Li Li [1], Lianqi Zhu [2],\*, Nan Xu [3], Ying Liang [1], Zhengyu Zhang [4], Junjie Liu [5] and Xin Li [6]**

1    School of Government, Beijing Normal University, Beijing 100875, China
2    College of Geography and Environmental Science, Henan University, Kaifeng 475004, China
3    School of Earth Sciences and Engineering, Hohai University, Nanjing 210098, China
4    School of Public Administration, China University of Geosciences, Wuhan 430074, China
5    State Key Laboratory of Resources and Environmental Information System, Institute of Geographic Sciences and Natural Resources Research, CAS, Beijing 100101, China
6    School of Architecture, Nanyang Institute of Technology, Nanyang 473004, China
*    Correspondence: lqzhu@henu.edu.cn

**Abstract:** Since land use/cover change profoundly impacts climate change and global warming has become an irreversible trend in the Anthropocene, there have been numerous global studies on the impact of climate change on vegetation growth (VG). However, the effects of climate extremes on the growth and direction of various vegetation types need to be better investigated, especially in the climate transition zones. In this paper, we examined the effect of diurnal warming on the growth of various types of vegetation in China's north–south transition zone. Based on the daily observation data of 92 meteorological stations in the Qinling-Daba (Qinba) mountainous area from 1982 to 2015, coupled with the Normalized Difference Vegetation Index (NDVI) and data on the type of vegetation. This research examined the temporal changes in the highest and lowest temperatures during the last 33 years using trend analysis. Second-order correlation analysis was used to investigate vegetation NDVI response characteristics to diurnal warming and to examine the effect of diurnal warming on the growth of different vegetation types. Our results showed that maximum temperature ($T_{max}$) and minimum temperature ($T_{min}$) showed an obvious upward trend, with the daytime temperature increase rate 1.2 times that at night, but failing the *t*-test. In addition, diurnal warming promoted vegetation growth, with NDVI associated positively correlated with $T_{max}$ at approximately 91.2% of the sites and 3492 rasters and with $T_{min}$ at roughly 53.25% of the sites and 2864 rasters. Spatial significance analysis showed an apparent difference, but few areas passed the *t*-test. Furthermore, daytime warming enhanced the growth of grasses, shrubs, deciduous broad-leaved forests, crops, and conifers, while the effect of nighttime warming on VG had a positive effect only on the growth of evergreen broad-leaved forest vegetation. These findings reveal the mechanisms of the impact of climate extremes on VG under global change, particularly the extent to which different vegetation types in climatic transitional zones respond to climate extremes.

**Keywords:** climate change; day and night temperature; vegetation NDVI; partial correlation analysis; Qinling-Daba mountains




## 1. Introduction

According to the fourth report of the Intergovernmental Panel on Climate Change (IPCC), the global average surface temperature increased by 0.74 °C between 1906 and 2005 and 0.65 °C over the past 50 years, with the latter warming at a rate that is about twice as fast as the former [1]. China's temperature has increased between 0.5 and 0.8 °C over the past century, and precipitation has varied widely [2]. The impacts of climate change on global terrestrial ecosystems are already significant. Vegetation is a natural link between

the various cycles and an essential part of terrestrial ecosystems [3,4]. The mechanism of climate influence on vegetation growth (VG) has become a crucial part of global change research [5,6]. In the 20th century, the climate-driven potential net primary productivity (NPP) of vegetation increased by 13% on a global scale [7]. Increasing precipitation increases VG in arid areas on regional scale [8], while it inhibits vegetation growth in humid and semi-humid regions [9,10]. Consequently, investigating the spatial nonstationary relationship between vegetation activity and climate change at varying spatial scales and its response mechanism has become a frontier area of research. Identifying the relationship between the influence of different climate factors on VG can help clarify the mechanism of global change in plant growth.

Previous studies have focused on measuring the interrelationships between VG and precipitation, temperature, and human activities [11–14]. In terms of vegetation–climate relationships, through an analysis of vegetation cover in Africa, Ghebrezgaber et al. [15] revealed that increased precipitation promotes VG, whereas increased temperature inhibits VG. Zhou et al. [16] found that droughts triggered by rising temperatures in the Northern Hemisphere were the main reason for the decline in vegetation cover at high latitudes during the 1980s. Zhao et al. [17] showed that precipitation was the main control factor affecting vegetation cover changes in China, and local climate conditions were more pronounced for VG in climate-affected areas. Piao et al. [18] concluded that the increase in temperature promoted the growth in NDVI, whereas the effect of precipitation on VG was more significant on a regional scale. In terms of VG and human activities, anthropogenic disturbances have dual effects on vegetation growth. On the one hand, rapid urbanization results in encroachment of construction land on agricultural land and forest land, resulting in a reduction in vegetation cover [19]. On the other hand, implementing environmental projects, such as returning farmland to forest and grass, is beneficial for enhancing vegetation cover [20]. Both climate change and anthropogenic impacts can have some effect on VG, and there are significant regional differences in the degree to which VG responds to climate change and anthropogenic impacts [21].

Since the beginning of the Anthropocene, the frequency and intensity of extreme climate events have increased significantly, posing a serious threat to the safety of human life and VG [22,23]. Academic studies have concentrated more on the interrelationship between VG and climate change [24,25], but the impact of climate extremes on VG has received less attention [26,27].

The Normalized Difference Vegetation Index (NDVI) is an indicator for monitoring and signaling changes in vegetation activity and productivity [28]. Its value reflects the level of vegetation activity [29]. As various NDVI products exist, the differences in sensors, spectral response function and correction methods, the accuracy of the study results is affected to some extent [30,31]. Monitoring the research of long-term vegetation dynamics and their impact mechanisms needs the selection of suitable NDVI data. In comparison to other NDVI data products, GIMMIS NDVI3g data have been widely used to monitor vegetation dynamics [32], land degradation [33], and carbon balance [34] on large regional scales because of their long monitoring time and high stability.

In addition, the magnitude of response to diurnal warming varies among vegetation types. In temperate regions, grass and scrub are more sensitive to summer daytime warming, whereas autumn warming significantly impacts broad-leaved and coniferous forests [35]. Rossi et al. [36] found that nighttime warming was more likely than daytime warming to promote earlier germination in black spruce. Zhao et al. [37] demonstrated, through the analysis of the mechanisms of VG response to diurnal warming in Xinjiang, that daytime warming was beneficial to the growth of coniferous forests, while nighttime warming had significant positive effects on coniferous forests, agricultural vegetation, and grassland. Despite the global research on the mechanisms of diurnal warming on the growth of various types of vegetation, there are few studies on the mechanisms of diurnal warming on typical vegetation types in climate-vegetation transitional areas, especially in the transitional zone from the northern subtropics to the warm temperate zone.

This study's main objective was to investigate the mechanism of diurnal warming on the growth of typical vegetation types in China's north–south transitional zone. We had two specific questions: (1) How does diurnal warming promote or inhibit vegetation growth at the site and raster scales? (2) How does diurnal warming affect the growth of typical vegetation types (grass, scrub, deciduous broad-leaved forest, evergreen deciduous broad-leaved mixed forest, evergreen broad-leaved forest, coniferous forest, and crops), and are there any differences in the response of various vegetation types to diurnal warming? First, we collected GIMMIS NDVI data, meteorological station data, and maps of vegetation type. The precipitation and daily minimum/maximum temperature datasets at the raster scale were formed by ANSUPLIN interpolation. Second, we measured the degree and significance of diurnal warming on vegetation NDVI at the site scale using second-order partial correlation. After controlling for precipitation variables, we further identify the effects of daily maximum temperature ($T_{max}$) and daily minimum temperatures ($T_{min}$) on typical vegetation. Finally, based on the interpolated climate data and NDVI, we applied second-order partial correlation to measure the degree that diurnal warming at the raster scale affects the NDVI of different vegetation types.

## 2. Data and Methods

### 2.1. Study Area

The Qinba Mountains (102°24′–112°40′ E, 30°43′–35°29′ N) are in the transitional zone between the north subtropical zone and the south warm temperate zone in China (Figure 1). They cover an area of about $3 \times 10^5$ km$^2$, including Hubei, Henan, Chongqing, Shaanxi, Gansu, and Sichuan provinces and cities. The climate types in the region are complex and diverse, with a north subtropical maritime climate, subtropical monsoon climate, temperate monsoon climate, and warm temperate continental climate. Climate has typical vertical variation on the growth of vegetation. In terms of topography, it is high in the northwest and low in the southeast, with hills, basins, valleys, and plains dominating the terrain. The eastern part of the study area is dominated by plains and hills, with an average elevation of 400 m above sea level. Its western part is composed of basins and valleys, with an average elevation of 1600 m. The spatial and temporal distribution of precipitation is uneven, due to monsoonal and continental climate, with an average annual precipitation of 450–1300 mm. The Qinba Mountains, located across the Yangtze River, Yellow River, and Huaihe River basins, boast well-developed water systems, abundant runoff, and 53% forest coverage, and hence are an important national biodiversity and water-conserving ecological function area.

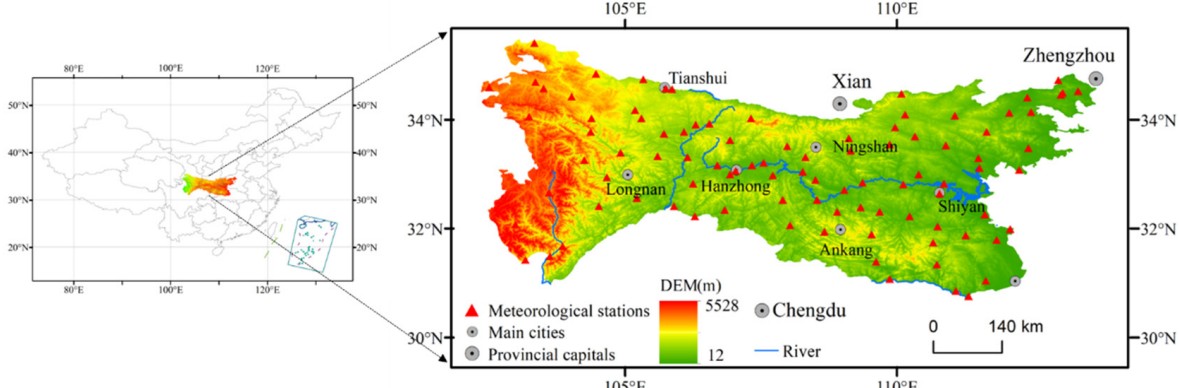

**Figure 1.** Distribution of meteorological stations in the Qinba Mountains.

### 2.2. Data Source and Processing

Vegetation data were mainly collected from the European Space Agency Climate Change Initiative Land Cover Website (ESA CCL-LC). We extracted the vegetation type in the Qinba Mountains changing from 1982 to 2015 and obtained the vegetation type shift and

change area during the study period using the Intersect tool in ArcGIS 10.3 [38]. It combines types of urban and bare land and waterbodies into one category and is not analyzed in this paper. The reclassification tool was used to calculate the areas where the vegetation types had not changed and to measure the dynamic growth of vegetation in response to diurnal warming of different vegetation types separately (Figure 2). The GIMMS3gV1.0 data from 1982 to 2015 were supplied by Global Inventory Modeling and Mapping Studies (GIMMS), USA, with a spatial resolution of 8 km, and NDVI datasets were synthesized every 15 d (http://www.resdc.cn/) (accessed on 16 August 2019). The meteorological data of $T_{max}$, $T_{min}$, and precipitation for 94 meteorological stations in the Qinba Mountains from 1982 to 2015 were collected from the China Meteorological Science Data Sharing Service (http://cdc.nmic.cn/) (accessed on 23 August 2019). The extreme temperatures ($T_{max}$ and $T_{min}$) were filtered from the daily weather station data, and then the extreme temperatures on the monthly and annual scales were obtained for each weather station. In order to determine whether there was a significant difference in the diurnal warming rates, we used analysis of covariance to calculate the correlation between the slopes of daytime and nighttime warming changes from 1982 to 2017 [39]. The digital elevation model (DEM) was derived from the geospatial data cloud (http://www.gscloud.cn/) (accessed on 24 August 2019) with a resolution of 30 m. It was downloaded from the United Stated Geological Survey website (USGS) (https://earthexplorer.usgs.gov/) (accessed on 4 September 2019) using a chunked download and was stitched and cropped to get the complete Qinba Mountain DEM data. On the basis of comprehensive consideration of the influence of terrain and number of stations on meteorological elements, the thin disk smooth spline function built into ANUSPLIN software uses smooth parameters for multivariate smooth interpolation of irregularly distributed data to achieve an optimal balance between data fidelity and smoothness of the fitted surface [40,41]. Currently, ANUSPLIN has been widely used for spatial interpolation of temperature and precipitation, and it has been shown that ANUSPLIN interpolation results are significantly better than other interpolation methods [42]. ANUSPLIN interpolation was used to interpolate the temperature and precipitation at each station into a 1 km × 1 km raster, which was resampled into an 8 km × 8 km raster [43]. Second-order partial correlation analysis, which refers to the calculation of the correlation coefficient between two variables when they are simultaneously correlated with a third variable by excluding the effect of the third variable, has been widely used in the study of global change [44,45]. Hence, the second-order partial correlation analysis was used to compute the correlation coefficients and significance levels between diurnal warming and NDVI of different vegetation types at the raster level.

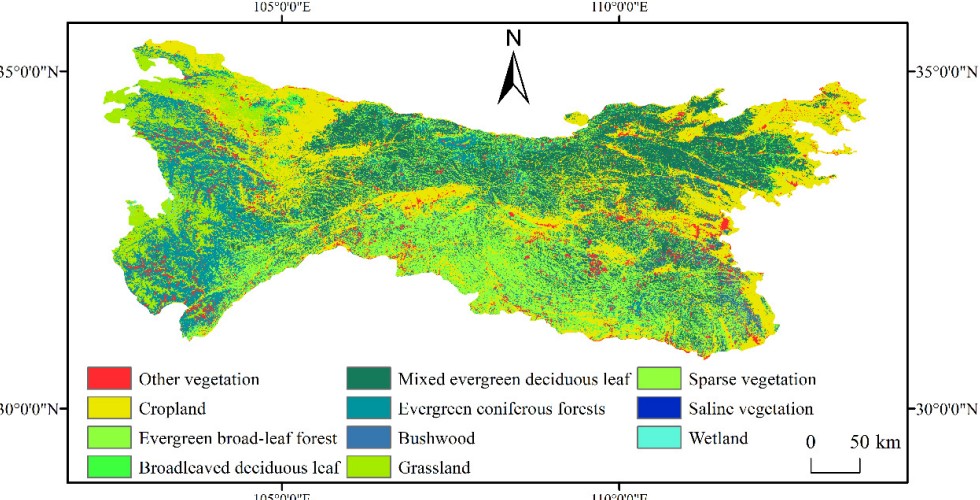

**Figure 2.** Vegetation-type distribution in the Qinling-Daba mountains from 1982 to 2015.

*2.3. Methodology*

In the Qinba Mountains, the response mechanism of transitional vegetation growth to diurnal warming has been a complex research topic. In this study, the effects of climate extremes on the growth of different vegetation types were thoroughly analyzed on two scales, site and raster, using multiple data sources, including NDVI, vegetation type, climate extreme indicators and DEM. In most previous research, the impact of climate change on VG was analyzed using a single-scale site or raster scale, which needs to be more precise. Here, we investigated the effect of climate extremes on the growth of various types at the site and grid scale, which can be used to verify and improve the results' reliability (Figure 3). In addition, we increased the accuracy of vegetation types by extracting areas with constant vegetation types between 1982 and 2015, which made up for the previous defects of single-year vegetation types.

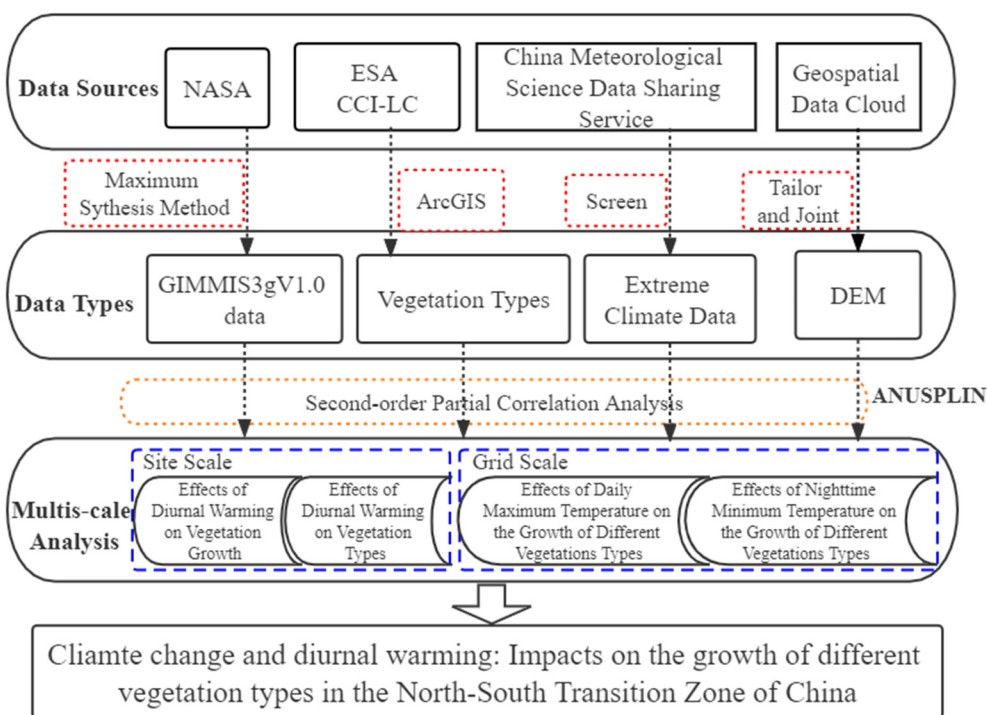

**Figure 3.** Research framework.

2.3.1. Maximum Value Composite Method

The maximum value of monthly NDVI is obtained through the maximum value synthesis (MVC) method, which can effectively reduce clouds, aerosols, cloud shadows, observation angles and solar altitude angles in the atmosphere [46]. The MVC method can be expressed as follows:

$$MNDVI_i = Max(NDVI_1, \ NDVI_2) \tag{1}$$

where *i* is the month serial number with values between [1,12]; $MNDV_i$ is the *NDVI* value of month *i*; $NDVI_1$ is the *NDVI* value for the first half of month *i*; $NDVI_2$ is the *NDVI* value for the second half of month *i*. The annual NDVI values are further calculated by calculating the monthly NDVI maxima [47].

2.3.2. ANUSPLIN Interpolation Method

ANUSPLIN interpolation is based on the theory of interpolation of ordinary thin disk and local thin disk spline functions, where the local thin disk smooth spline method is an extension of the thin disk smooth spline prototype, which allows the introduction of

linear covariates to the model except for the ordinary spline independent variables also allows the introduction of linear covariate sub-models [48,49]. The interpolation software ANUSPLIN is capable of interpolating multiple surfaces simultaneously and has a set of detailed statistical analysis, data processing and calculation of standard errors of spatial distribution, with the advantages of high computational efficiency and ease of operation [50]. Therefore, we used DEM as a covariate to interpolate $T_{max}$, $T_{min}$ and precipitation in the Qinba Mountains by ANUSPLIN software. The ANUSPLIN interpolation method can be expressed as follows:

$$z_i = f(x_i) + b^T y_i + e_i \quad (i = 1, 2, 3, \ldots, N) \tag{2}$$

where $z_i$ is the dependent variable located at spatial location; $f$ is an unknown smooth function about $x_i$; $x_i$ is a $d$-dimensional vector with respect to the sample independent variables; $b$ is the $p$ maintenance of $y_i$; $y_i$ is the $p$ dimensional independent covariates; $T$ is the number of iterations; $e_i$ is the random error of the independent variable with expectation 0. The function $f$ and the vector of coefficient $b$ can be calculated as follows:

$$\sum_{i=1}^{N} \left( \frac{z_i - f(x_i) - b^T y_i}{w_i} \right) + \rho J_m(f) \tag{3}$$

where $w_i$ is the known local relative coefficient of variation used as a weight; $\rho$ is the smooth parameter, determined by the Bayesian-based generalized maximum likelihood method provided by ANUSPLIN [51]; $J_m$ $(f)$ is the m-order partial derivative of the function $f$.

### 2.3.3. Second-Order Partial Correlation Analysis Method

Partial correlation analysis measures the linear correlation of two variables by removing the linear effects of other variables. The first-order partial correlation coefficient is the result obtained by excluding the effect of one variable, the second-order partial correlation coefficient is the result obtained by excluding the effect of two variables, and the correlation coefficient is the result obtained when no other variables are excluded [52]. In this study, second-order partial correlation analysis is used because three climatic factors ($T_{max}$, $T_{min}$ and precipitation) are employed. The effects of precipitation and $T_{min}$ ($T_{max}$) need to be excluded when the effects of $T_{max}$ ($T_{min}$) on the NDVI of vegetation are analyzed. The correlation coefficients are calculated as follows:

$$r_{xy} = \frac{\sum\limits_{i=1}^{n} (x_i - x_{mean})(y_i - y_{mean})}{\sqrt{\sum_{i=1}^{n}(x_i - x_{mean})\sum_{i=1}^{n}(y_i - y_{mean})}} \tag{4}$$

where $x$ is the NDVI of vegetation; $y$ is the $T_{max}$ ($T_{min}$); $x_{mean}$ is the average of elements NDVI; $y_{mean}$ is the average of elements $T_{max}$ ($T_{min}$); $r_{xy}$ is the correlation coefficient of NDVI and $T_{max}$ ($T_{min}$). The first-order partial correlation analysis coefficient is calculated as:

$$r_{xy-z} = \frac{r_{xy} - r_{xz}r_{yz}}{\sqrt{1 - r_{xz}^2}\sqrt{1 - r_{yz}^2}} \tag{5}$$

The second-order partial correlation analysis coefficient is calculated as follows:

$$r_{xy-z-u} = \frac{r_{xy-z} - r_{xu-z}r_{yu-z}}{\sqrt{1 - r_{xu-z}^2}\sqrt{1 - r_{yu-z}^2}} \tag{6}$$

where $r_{xy-z}$ is the bias correlation coefficient between NDVI and $T_{max}$ ($T_{min}$) excluding variable $z$; $z$ is the precipitation; $r_{xz}$ is the NDVI and precipitation correlation coefficient; $r_{yz}$ is the $T_{max}$ ($T_{min}$) and precipitation correlation coefficient; $r_{xy-z-u}$ is the bias correlation coefficient of NDVI and $T_{max}$ ($T_{min}$) excluding variables precipitation and $T_{min}$ ($T_{max}$); $u$

is the $T_{max}$ ($T_{min}$). The *t*-test is used to test the significance of the second-order partial correlation coefficients and is calculated as follows:

$$t = \frac{r\sqrt{n - q - 1}}{\sqrt{1 - r^2}} \tag{7}$$

where *r* is the partial correlation coefficient; *n* is the number of samples; *q* is the number of degrees of freedom. Referring to previous studies [53] and based on the actual situation, the correlation coefficient between vegetation NDVI and day–night temperature increase was divided into moderate positive correlation ($0.5 \leq r < 0.8$), low positive correlation ($0.3 \leq r < 0.5$), weak positive correlation ($0 \leq r < 0.3$), weak negative correlation ($-0.3 \leq r < 0$), and low negative correlation ($-0.5 \leq r < -0.3$) and moderate negative correlation ($-0.8 \leq r < -0.5$).

## 3. Results

### 3.1. Analysis of Diurnal Warming Trends in the Qinba Mountains

There was a significant upward trend in the $T_{max}$ and $T_{min}$ in the Qinba Mountains between 1982 and 2015. Figure 4 indicates that the variation of $T_{max}$ was higher than $T_{min}$, where $T_{max}$ increased 0.5 °C per decade and $T_{min}$ increased 0.4 °C per decade.

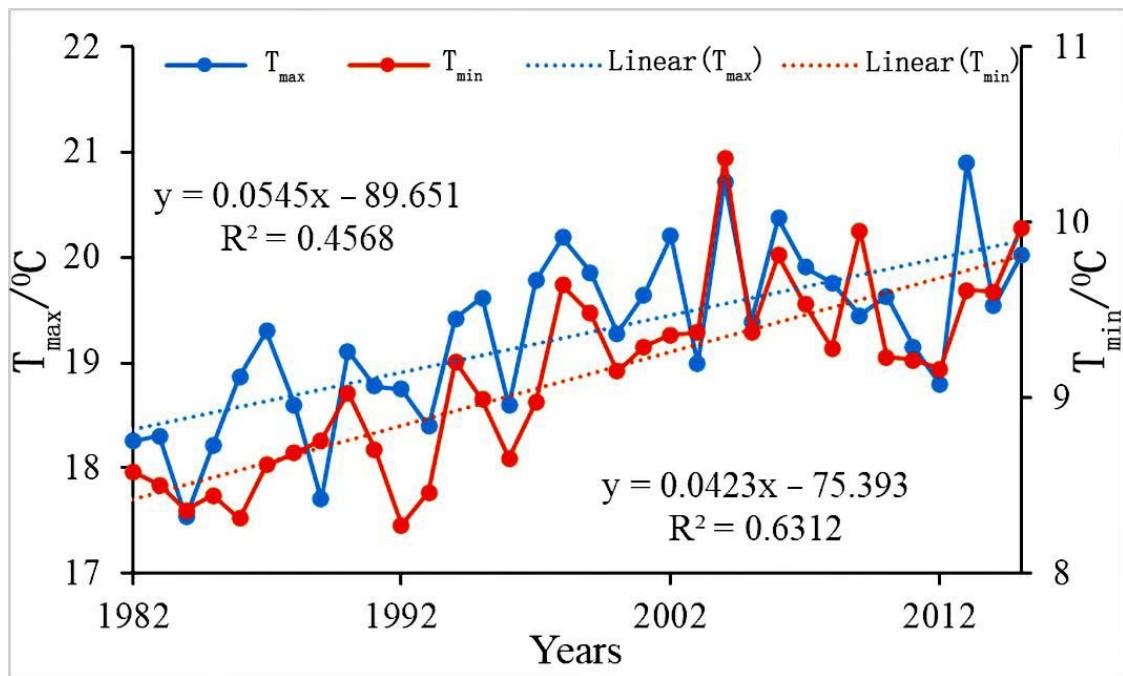

**Figure 4.** Variation trends of $T_{max}$ and $T_{min}$ in Qinba Mountains from 1982 to 2015.

### 3.2. Effect of Site-Scale Diurnal Warming on Different Vegetation Types in the Qinba Mountains

3.2.1. Significance Analysis of Diurnal Warming and NDVI Activity on the Site Scale

The GIMMIS sensor NDVI and $T_{max}$ bias second-order correlation are computed in this paper, by excluding the effects of precipitation and $T_{min}$ (Figure 5). About 91.2% of the 92 meteorological stations in the Qinba Mountains showed a positive correlation, of which about 24.09% passed the *t*-test ($p < 0.01$) and 21.68% of the stations passed the *t*-test ($p < 0.05$). The percentage of meteorological stations that passed the negative correlation test was 8.7%, among which about 12.5% displayed a negative correlation ($p < 0.05$). After removing the effects of precipitation and $T_{max}$, about 53.25% of the meteorological stations showed a positive correlation between NDVI and $T_{min}$. Where about 2.04% passed the *t*-test ($p < 0.01$) and about 20.4% of the meteorological stations were positively correlated

($p < 0.05$). Similarly, about 46.75% of the meteorological stations displayed a negative correlation between vegetation NDVI and $T_{min}$, with about 9.3% showing a negative correlation ($p < 0.01$) and about 2.3% with a negative correlation ($p < 0.05$). The stations where NDVI and $T_{max}$ passed the *t*-test were relatively large and concentrated in the middle- and low-elevation regions. In contrast, the sites where NDVI and $T_{min}$ passed the *t*-test were scattered at the edges of the Qinba Mountains, such as in the northwest, northeast, and southeast.

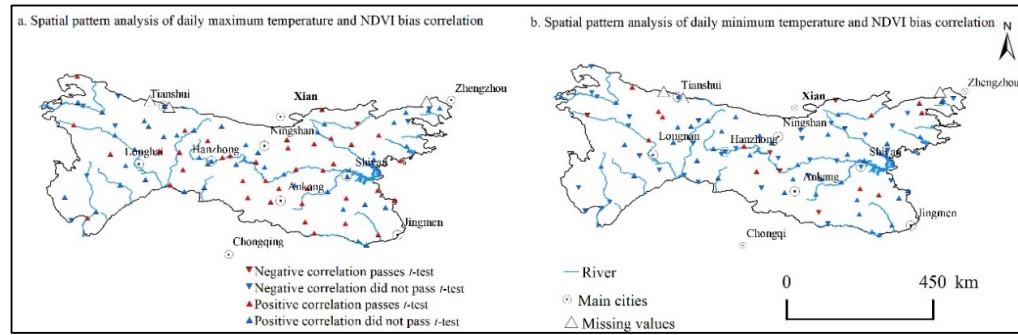

**Figure 5.** Second-order partial correlation analysis of NDVI with $T_{max}$ (**a**) and $T_{min}$ (**b**) in the Qinba Mountains from 1982 to 2015.

3.2.2. Correlation Analysis of Different Vegetation Types on Diurnal Warming

The $T_{max}$, $T_{min}$ and precipitation data of meteorological stations in the Qinba Mountains from 1982 to 2015 were extracted, and the second-order correlation coefficients and significance of each vegetation type with diurnal warming were calculated on the basis of NDVI data of various types of vegetation at each station. Differences in NDVI bias correlations for $T_{max}$ and $T_{min}$ were found for different types of vegetation (Table 1). In terms of the correlation between vegetation types and $T_{max}$, except for crops, which passed the negative correlation ($p < 0.01$), all other vegetation types were positively correlated with $T_{max}$. All other positive correlation values were below 0.2 and did not pass the *t*-test, except for mixed evergreen broad-leaved forests, which had a higher correlation of 0.414. In terms of the correlation between different vegetation types and $T_{min}$: coniferous forest, scrub, and crops all showed a positive correlation with $T_{min}$ ($p < 0.01$); grass showed a positive correlation with $T_{min}$, and evergreen broad-leaved forest and mixed evergreen broad-leaved forest showed negative correlation with $T_{min}$, but none of the above passed the *t*-test. Overall, daytime warming was found to promote the growth of mixed evergreen deciduous broad-leaved forests, grasses, evergreen broad-leaved forests, scrubs, and coniferous forests and inhibit the growth of crops. The effect of increasing nighttime temperature on VG was inhibited only in mixed deciduous broad-leaved and evergreen broad-leaved forests, but had a positive effect on all other vegetation types, which significantly promoted the growth of coniferous forests and scrub.

**Table 1.** Partial correlation analysis of different vegetation types with $T_{max}$, $T_{min}$ in the Qinba Mountains.

| Vegetation Type | $T_{max}$ | $T_{min}$ |
|---|---|---|
| Grass | 0.182 | 0.323 |
| Scrub | 0.156 | 0.776 ** |
| Deciduous broad-leaved forest | - | - |
| Evergreen deciduous broad-leaved mixed forest | 0.414 | −0.136 |
| Evergreen broad-leaved forest | 0.168 | −0.003 |
| Coniferous forest | 0.146 | 0.843 ** |
| Crops | −0.145 ** | 0.175 ** |

Note: ** indicate passing $p < 0.01$ *t*-tests.

*3.3. Effect of Grid-Scale Diurnal Warming on Different Vegetation Types in the Qinba Mountains*

3.3.1. Spatial Response Characteristics of NDVI to Daytime Maximum Temperatures in Different Vegetation Types

Figure 6 and Table 2 indicate that the direction and intensity of the effect of increasing daytime maximum temperature ($T_{max}$) on VG in the Qinba Mountains varied spatially. On the whole, the increase in $T_{max}$ positively affected all types of vegetation. Low positive and weak positive correlations dominated the correlation between NDVI and $T_{max}$. The moderate positive correlation was clustered in the middle of the study area in a band, which was consistent with the spatial pattern of the *t*-test, suggesting that the region where $T_{max}$ promoted VG passed the *t*-test ($p < 0.01$). Few areas within the grass, scrub and cultivated plant types passed the overall *t*-test ($p < 0.05$), revealing the varying degrees of sensitivity and regional variability of different vegetation types to $T_{max}$. According to the correlation coefficient rank, the primary correlations between vegetation types and $T_{max}$ are low positive correlation, weak positive correlation, and weak negative correlation. The number of grids with moderately positive correlation had a high proportion of 103, 87 and 98, respectively, and all of them passed the *t*-test ($p < 0.01$), indicating that $T_{max}$ promotes the growth of vegetation types such as evergreen broad-leaved forest, evergreen deciduous broad-leaved forest and crops with significant intensity. Other vegetation types were covered by fewer grids that passed the *t*-test ($p < 0.05$) and moderately positively correlated. In terms of low positive correlation, mixed evergreen deciduous broad-leaved forests accounted for the highest percentage (35.83%), with 63 grids passing the *t*-test ($p < 0.01$) and 163 significant test grids accounting for 18.81% and 48.66%, respectively. Evergreen broad-leaved forests were the second-most important, with 314 grids and 32.50% of the weight, and 197 grids passing the *t*-test, accounting for 62% of the low positive correlation. Low positive correlation grasslands included the smallest proportion of rasters, 10.61%, and deciduous broad-leaved forests comprised the fewest rasters, with 19 images. Regarding the level of weak positive correlation, grasses occupied 226 rasters, with a weight of 68.48%, while 19 pixels passed the *t*-test, with a weight of 8.41%. The results of the research demonstrated that there were differences in the degree of response of different vegetation types to daytime warming. The increase in $T_{max}$ promoted VG in general, and the reaction of evergreen broad-leaved forests, mixed evergreen deciduous broad-leaved forests, coniferous forests, and crops was more pronounced. The weak positive correlation between deciduous broad-leaved forests and grasses and the $T_{max}$ increase reflects the weak effect of daytime warming on deciduous broad-leaved forests and grasses.

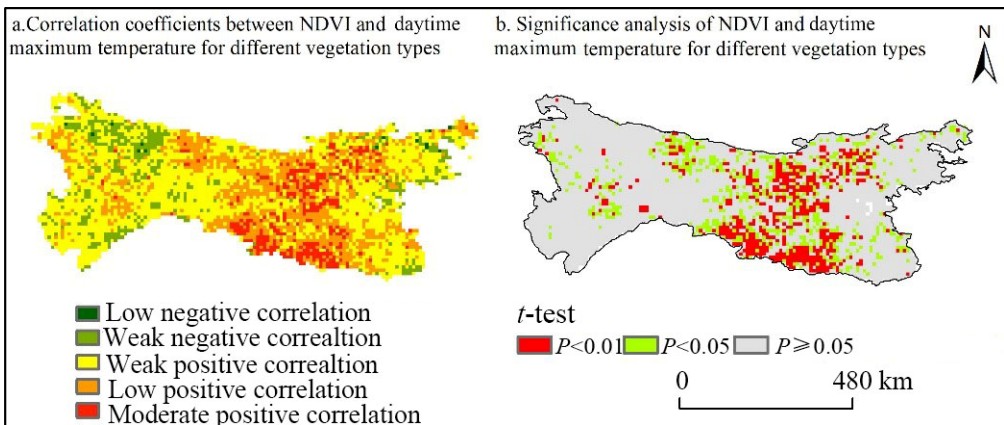

**Figure 6.** Correlation coefficient (**a**) and *t*-test analysis (**b**) between NDVI and daytime maximum temperature ($T_{max}$) of different vegetation types from 1982 to 2015.

**Table 2.** Numbers and *t*-tests of grids with different correlation coefficients between different vegetation types and $T_{max}$ in Qinba Mountains.

| Correlation Coefficient Grade | | Grass | | Scrub | | Deciduous Broad-Leaved Forest | | Evergreen Deciduous Broad-Leaved Mixed Forest | | Evergreen Broad-Leaved Forest | | Coniferous Forest | | Crops | |
|---|---|---|---|---|---|---|---|---|---|---|---|---|---|---|---|
| | | Number | % | Number | % | Number | % | Number | % | Number | % | Number | % | Number | % |
| Moderately positive correlation | $T_{max}$ | - | - | 20 | 12.2 | 6 | 7.5 | 87 | 9.30 | 103 | 11.55 | 15 | 3.10 | 98 | 9.21 |
| | $p < 0.01$ | - | - | 20 | 100 | 6 | | 87 | 100 | 103 | 100 | 15 | 100 | 98 | 100 |
| | $p < 0.05$ | - | - | - | - | - | - | - | - | - | - | - | - | - | - |
| Low positive correlation | $T_{max}$ | 35 | 10.61 | 54 | 32.93 | 19 | 23.75 | 335 | 35.83 | 314 | 35.20 | 107 | 22.11 | 234 | 21.99 |
| | $p < 0.01$ | - | - | 3 | 5.56 | - | - | 63 | 18.81 | 34 | 10.83 | 8 | 7.48 | 38 | 16.24 |
| | $p < 0.05$ | - | - | 31 | 57.41 | 11 | 57.89 | 163 | 48.66 | 163 | 51.91 | 55 | 51.40 | 121 | 51.71 |
| Weak positive correlation | $T_{max}$ | 226 | 68.48 | 79 | 48.17 | 44 | 55.00 | 468 | 50.05 | 432 | 48.43 | 304 | 62.81 | 512 | 48.12 |
| | $p < 0.01$ | 3 | 1.33 | - | - | - | - | - | - | - | - | - | - | - | - |
| | $p < 0.05$ | 16 | 7.08 | - | - | - | - | - | - | - | - | - | - | - | - |
| Weak negative correlation | $T_{max}$ | 69 | 20.91 | 11 | 6.70 | 11 | 13.75 | 44 | 4.71 | 43 | 4.82 | 58 | 11.98 | 209 | 19.64 |
| | $p < 0.01$ | - | - | - | - | - | - | - | - | - | - | - | - | - | - |
| | $p < 0.05$ | - | - | - | - | - | - | - | - | - | - | - | - | - | - |
| Low negative correlation | $T_{max}$ | - | - | - | - | - | - | 1 | 0.11 | - | - | - | - | 11 | 1.03 |
| | $p < 0.01$ | - | - | - | - | - | - | - | - | - | - | - | - | 1 | 9.09 |
| | $p < 0.05$ | - | - | - | - | - | - | - | - | - | - | - | - | 1 | 9.09 |
| Moderately negative correlation | $T_{max}$ | - | - | - | - | - | - | - | - | - | - | - | | - | - |
| | $p < 0.01$ | - | - | - | - | - | - | - | - | - | - | - | - | - | - |
| | $p < 0.05$ | - | - | - | - | - | - | - | - | - | - | - | | - | - |
| Total | | 330 | 100 | 164 | 100 | 80 | 100 | 935 | 100 | 892 | 100 | 484 | | 1064 | |

### 3.3.2. Spatial Response Characteristics of NDVI to Nighttime Minimum Temperatures in Different Vegetation Types

Figure 7 indicates that the effect of nighttime minimum temperatures ($T_{min}$) on VG was insignificant, dominated by weak positive and weak negative correlations (Table 3). Moderately positive correlations were concentrated in the northwest, southeast, and at the border, consistent with the spatial pattern that passed the *t*-test ($p < 0.01$), indicating that the area where $T_{min}$ promoted VG was highly significant. Regional variability existed between VG types for $T_{min}$, with evergreen deciduous broad-leaved mixed forest exhibiting a higher sensitivity to nighttime warming: the number of image elements in positive correlation was 702, accounting for 75.08%, and the raster number passing the *t*-test was 114, accounting for 16.24%; raster numbers in low negative correlation and weak negative correlation were 18 and 215, respectively, and only the number of 60 pixels in the weak negative correlation passed the *t*-test. Concerning the weak positive correlation, the image elements proportion accounted for by different vegetation types ranged from 39.57% to 53.70%, with the highest proportion (53.70%), with evergreen broad-leaved forests with crops possessing the lowest percentage (39.57%), and only the presence of image elements in coniferous forests passed the *t*-test. Regarding the low positive correlation, nighttime warming had an enhanced effect on VG promotion, as a higher number of rasters passing the *t*-test ($p < 0.05$). More than half the grids passed the *t*-test for both crops and broad-leaved evergreen forests—201 and 106, respectively—demonstrating that the increase in $T_{min}$ significantly increased the growth rate of vegetation. In terms of moderately positive correlation, the number of rasters accounted by the vegetation types was small, and only coniferous forests failed the *t*-test, reflecting that the stronger the promotion effect of $T_{min}$ on VG, the higher the degree of significance. In terms of negative correlations, weak negative correlations were the primary type of $T_{min}$ on VG in different regions. Only crops and grasses had rasters with a moderate negative correlation and passed the *t*-test, suggesting that the increase in $T_{min}$ had a more significant suppressive effect on crops and grasses. The rasters of broad-leaved evergreen forests, crops, and grasses passed the *t*-test. Both mixed evergreen deciduous broad-leaved and coniferous forests had grids that passed *t*-test on weak negative correlation with a weight of 27.91% and 9.23%, respectively. It shows that nighttime warming inhibits evergreen broad-leaved forests, mixed evergreen deciduous broad-leaved forests, crops, and grasses more than other vegetation types.

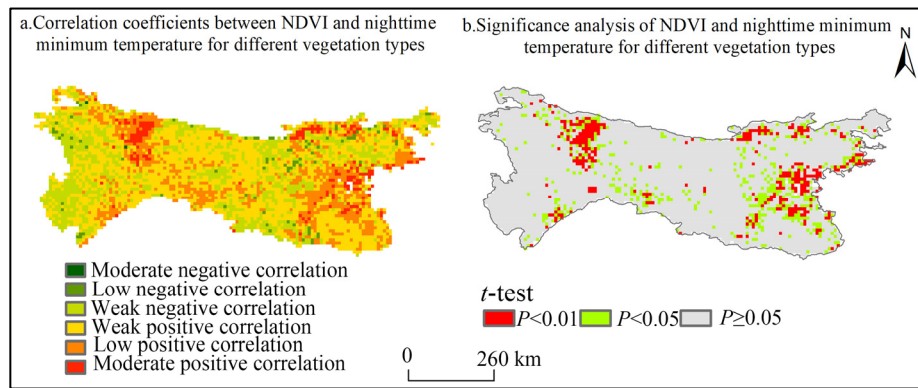

**Figure 7.** Correlations (**a**) and *t*-test analysis (**b**) between NDVI and nighttime minimum temperatures ($T_{min}$) of different vegetation types from 1982 to 2015.

**Table 3.** Numbers and *t*-tests of grids occupied by different vegetation types and different $T_{min}$ correlation coefficients in Qinba Mountains.

| Correlation Coefficient Grade | | Grass | | Scrub | | Deciduous Broad-Leaved Forest | | Evergreen Deciduous Broad-Leaved Mixed Forest | | Evergreen Broad-Leaved Forest | | Coniferous Forest | | Crops | |
|---|---|---|---|---|---|---|---|---|---|---|---|---|---|---|---|
| | | Number | % | Number | % | Number | % | Number | % | Number | % | Number | % | Number | % |
| Moderately positive correlation | $T_{min}$ | 2 | 0.61 | 6 | 3.66 | 5 | 6.25 | 39 | 4.17 | 12 | 1.34 | 8 | 1.65 | 103 | 9.68 |
| | $p < 0.01$ | 2 | 1 | 6 | 100 | 5 | 100 | 3 | 7.69 | 12 | 100 | - | - | 103 | 100 |
| | $p < 0.05$ | - | - | - | - | - | - | 6 | 15.38 | - | - | - | - | - | - |
| Low positive correlation | $T_{min}$ | 39 | 11.82 | 41 | 25 | 18 | 22.5 | 202 | 21.60 | 195 | 21.86 | 40 | 8.26 | 300 | 28.20 |
| | $p < 0.01$ | 6 | 15.38 | 6 | 14.63 | 2 | 11.11 | 24 | 11.88 | 11 | 5.64 | 4 | 10.00 | 57 | 19.00 |
| | $p < 0.05$ | 13 | 33.33 | 23 | 56.10 | 9 | 50 | 15 | 7.43 | 95 | 48.72 | 2 | 5.00 | 144 | 48.00 |
| Weak positive correlation | $T_{min}$ | 155 | 46.97 | 81 | 49.39 | 39 | 48.75 | 461 | 49.30 | 479 | 53.70 | 218 | 45.04 | 421 | 39.57 |
| | $p < 0.01$ | - | - | - | - | - | - | 26 | 5.64 | - | - | 4 | 1.83 | - | - |
| | $p < 0.05$ | - | - | - | - | - | - | 40 | 8.68 | - | - | 11 | 5.05 | - | - |
| Weak negative correlation | $T_{min}$ | 130 | 39.39 | 31 | 18.90 | 16 | 20 | 215 | 22.99 | 188 | 21.08 | 206 | 42.56 | 216 | 20.30 |
| | $p < 0.01$ | - | - | - | - | - | - | 19 | 8.84 | - | - | 5 | 2.43 | - | - |
| | $p < 0.05$ | - | - | - | - | - | - | 41 | 19.07 | - | - | 14 | 6.80 | - | - |
| Low negative correlation | $T_{min}$ | 3 | 0.91 | 5 | 3.05 | 2 | 2.5 | 18 | 1.93 | 18 | 2.92 | 12 | 2.48 | 23 | 2.16 |
| | $p < 0.01$ | - | - | 3 | 60.00 | - | - | - | - | 1 | 5.56 | - | - | 2 | 8.70 |
| | $p < 0.05$ | 1 | 33.33 | - | - | - | - | - | - | 4 | 22.22 | - | - | 1 | 4.35 |
| Moderately negative correlation | $T_{min}$ | 1 | 0.30 | - | - | - | - | - | - | - | - | - | - | 1 | 0.09 |
| | $p < 0.01$ | 1 | 1 | - | - | - | - | - | - | - | - | - | - | 1 | 100 |
| | $p < 0.05$ | - | - | - | - | - | - | - | - | - | - | - | - | - | - |
| Total | | 330 | 100 | 164 | 100 | 80 | 100 | 935 | 100 | 892 | 100 | 484 | 100 | 1064 | 100 |

## 4. Discussion

Regarding the effect of diurnal warming on VG, the correlation between vegetation NDVI and $T_{max}$ and $T_{min}$ was analyzed using meteorological station data. The results showed that about half the meteorological stations passed the *t*-test with $T_{max}$, whereas few stations indicated a significant correlation with $T_{min}$, showing higher significance for daytime warming on vegetation growth at the station level. Additionally, about 91.2% of the sites were positively correlated with $T_{max}$ and 53.25% were positively correlated with $T_{min}$, demonstrating that VG in the Qinba Mountains is more significantly affected by daytime warming. This is consistent with Peng et al.'s [9] finding that increased daytime temperatures greatly promote VG and its ecosystem carbon sink function in boreal and temperate humid regions, but contrary to the results of Zhao et al. [37], who studied the effect of diurnal warming on VG in Xinjiang. Possible causes are the spatial heterogeneity of the region, the number of sites and the difference in vegetation type products [54,55].

On the other hand, ANUSPLIN was used to interpolate the station data to obtain meteorological data at the 8 km × 8 km raster level. The correlation and significance of NDVI and diurnal warming of vegetation on the raster scale were further analyzed, the result revealing that diurnal warming was the dominating factor in VG promotion. Both site and raster scales indicated that diurnal warming promoted VG. Still, the degree of significance was low and somewhat related to the number of meteorological sites and

the effect of ANUSPLIN interpolation accuracy. Additionally, the natural environment (soil quality, topography, parent material, etc.), climatic factors (temperature, precipitation, light intensity, sunshine number, extreme weather), and human activities, all influence VG, which may be one of the reasons affecting the *t*-test of diurnal warming and NDVI.

Correlations and *t*-test analysis between diurnal warming and different types of VG on both site and raster scales indicate that the positive effect of diurnal warming on VG is more widely distributed than the inhibitory effect. The impact of an increase in daytime temperature on VG is more significant, mainly because an increase in daytime temperature promotes the transpiration and respiration of vegetation while increasing the rate of $CO_2$ into the leaf area to enhance photosynthetic efficiency, thus promoting the accumulation of nutrients in vegetation [56]. Analysis of the *t*-test showed that daytime warming significantly promoted the growth of mixed evergreen deciduous broad-leaved forests, grasses, evergreen broad-leaved forests, scrub, and coniferous forests, while inhibiting the growth of crops. The effect of increasing nighttime temperature on VG was only inhibited in mixed evergreen deciduous broad-leaved and evergreen broad-leaved forests, but had a positive impact on all other vegetation types, among which the effect was more evident in coniferous forests and scrub. Nighttime warming had a positive impact on the growth of coniferous forests and other plants by regulating leaf carbohydrate content [57,58], reducing frost damage [59,60], and enhancing the resistance of plant communities to drought. In addition, nocturnal autotrophic respiration has a negative impact on the growth of mixed evergreen deciduous broad-leaved and evergreen broad-leaved forests by elevating the plant autotrophic respiration rate [61,62], shortening the plant filling period [63,64] and reducing the size of endosperm cells at the maturity stage [65], which results in diurnal warming. On the site scale, the effect of diurnal warming on deciduous broad-leaved forests was not analyzed, because deciduous broad-leaved forests were not distributed over the site when vegetation types were extracted.

Due to the changes in altitude and temperature, the ecosystem in mountainous regions is characterized by a complicated environmental gradient and vertical differentiation of forest vegetation. From the foot to the top of the hills, in proper order, are various vegetational forms such as evergreen broad-leaved forests, broadleaved deciduous forests and coniferous forests, which determine the relationship between climate extremes and vegetation in spatial pattern [66]. Moreover, the same vegetation influences the climate differently during various menstrual periods. For example, as the growing season of vegetation on the Qinghai–Tibet Plateau is positively affected by nighttime warming, revival comes earlier [67]. In the nongrowing season, however, nighttime warming causes the postponement of the early phenological phenomena of vegetation while daytime warming will cause an increase in the thermal buildup, resulting in the early appearance of these phenomena [68]. All these studies demonstrate that multiple factors influence vegetation growth and that there is an intricate nonlinear relationship between vegetation and the natural environment, climate change, and human activities.

This paper clarifies the mechanism of diurnal warming on VG in the transition zone and reveals the response mechanisms of different vegetation types to diurnal warming. The results of this study can be used as a reference for vegetation-climate studies in other climatic transition zones around the world, especially in the transition from subtropical to warm temperate zones. In future research, we could further develop the following: (1) to further investigate the NDVI of vegetation in the Qinba Mountains based on MODIS data, to monitor the dynamic growth of vegetation for a long time period by fusing from multiple sources and to study the effect of global changes on VG; (2) by simulating climate models, improve the accuracy of meteorological interpolation and vegetation types, and detect the effects of diurnal warming on the growth of different types of vegetation under future scenarios; (3) to screen representative extreme climate indicators and analyze in depth the impact of different extreme climates (precipitation, temperature) on VG, with a particular focus on monitoring the effect of major meteorological disasters on VG; and (4) give priority to the analysis of the mechanisms of influence of extreme climate on the phenological

phenomena of vegetation based on the collection of data from multiple sources, in order to clarify the responsiveness of the same vegetation to daytime and nighttime warming during different growing periods.

## 5. Conclusions

Global warming is an essential element of future climate change, and diurnal warming is already significantly impacting VG. In this study, we found that the daytime maximum temperature increase rate was 1.2 times higher than the daytime minimum temperature increase rate in the Qinba Mountains from 1982 to 2015.

Furthermore, diurnal warming had an overall positive effect on VG in the Qinba Mountains, with daytime warming being more effective than nighttime warming in promoting vegetation growth. Approximately 91.2% of meteorological stations had a positive correlation between $T_{max}$ and NDVI, with 45.27% passing the *t*-test. The proportion of all types of NDVI exhibiting a positive correlation with $T_{max}$ at the raster level exceeded 85%. About 53.25% of the meteorological stations passed the positive correlation test between NDVI and $T_{min}$. At the raster level, the correlations between $T_{min}$ and NDVI were insignificant, with primarily weak positive and weak negative correlations, indicating that daytime warming had more significant effects on VG.

Lastly, in terms of correlations and *t*-tests between different vegetation types and $T_{max}$ and $T_{min}$, daytime warming greatly promoted the growth of grasses, scrub, deciduous broad-leaved forests, crops and coniferous forests, and inhibited the growth of evergreen broad-leaved forests in the study area. The effect of nighttime warming on VG was positive only for the growth of evergreen broad-leaved forest vegetation and negative for other vegetation types.

**Author Contributions:** Conceptualization, L.L. and L.Z.; methodology, L.L.; software, L.L.; validation, N.X., Y.L. and Z.Z.; formal analysis, L.L.; investigation, L.Z.; resources, L.Z.; data curation, L.L.; writing—original draft preparation, J.L.; writing—review and editing, L.L.; visualization, X.L.; supervision, Z.Z. All authors have read and agreed to the published version of the manuscript.

**Funding:** The work presented in this paper was supported by the National Key Research and Development Program of China (grant 2021YFE0106700) and the National Science and Technology Basic Resource Investigation Program of China (grant 2017FY1009002).

**Institutional Review Board Statement:** Not applicable.

**Informed Consent Statement:** Not applicable.

**Data Availability Statement:** Not applicable.

**Conflicts of Interest:** The authors declare no conflict of interest.

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
