# Peer review of "Climate Change and Diurnal Warming: Impacts on the Growth of Different Vegetation Types in the North–South Transition Zone of China"

_land, doi:10.3390/land12010013_

Round 1
Reviewer 1 Report (Previous Reviewer 2)
The authors modified and enriched the manuscript in accordance with my recommendations.
Author Response
We would like to thank the editorial staff and the reviewers for their time in publishing this article, which has been revised and improved in detail based on the reviewers' comments. Once again, thank you for your hard work.

Reviewer 2 Report (Previous Reviewer 1)
The authors have improved their manuscript, but it still presents a number of scientific rigor issues that call into question its suitability for publication. These detected issues are:
1) The authors changed the word "asymmetry" into "inconsistency" in Section 3.1. This change is inconsistent with the rest of the manuscript and prior literature, which uses "asymmetry", and only circumvents the main issue. As I mentioned in the previous round of review, the authors need to formally reject the null hypothesis that the rate of increase in Tmax and Tmin are equal before making any mention of the difference (e.g., in the abstract: "the temperature increase rate in the daytime was 1.2 times that at night"). This is by no means a difficult check to make, but requires familiarity with the concept of confidence intervals for regression coefficients.
2) The authors have added sections 2.3.1 Maximum value composite method and 2.3.2 ANUSPLIN interpolation method, which provide useful methodological information. However, the section on ANUSPLIN is far from providing the details required for scientific reproducibility. Also, in both these new sections, prior to each equation, the authors use the wording "The correlation coefficients are calculated as follows:", despite none of the equations being even remotely related to correlation coefficients. Such careless revisions call into question the rigor put into the rest of the manuscript.
Author Response
Thank you for giving me the opportunity to submit a revised draft of my manuscript titled ‘Climate change and diurnal warming: Impacts on the growth of different vegetation types in the North-South Transition Zone of China’ (Manuscript ID: Land-2034471) to Land. We appreciate the time and effort that you and the reviewers dedicated to providing you valuable feedback on my manuscript. We are grateful to the reviewers for the insightful comments on my paper. All these comments were valuable and extremely helpful for revising and improving our paper. We have studied the comments carefully and have made corrections, which we hope will meet with your approval. We have highlighted the changes within the manuscript.
Here is a point-by-point response to the reviewers’ comments and concerns.
Point 1: The authors changed the word "asymmetry" into "inconsistency" in Section 3.1. This change is inconsistent with the rest of the manuscript and prior literature, which uses "asymmetry", and only circumvents the main issue. As I mentioned in the previous round of review, the authors need to formally reject the null hypothesis that the rate of increase in Tmax and Tmin are equal before making any mention of the difference (e.g., in the abstract: "the temperature increase rate in the daytime was 1.2 times that at night"). This is by no means a difficult check to make, but requires familiarity with the concept of confidence intervals for regression coefficients.
Response 1: Thank you for your comments. We think your opinion is very reasonable. We apologize for not responding well to your comments in the last revision. An introduction to diurnal asymmetric warming has been added to our introduction, and references have been introduced for clarification. In addition, the asymmetry in diurnal warming was further analyzed according to the reviewer's comments, and we used the analysis of covariance method in R software to compare whether there was a significant difference between the slope of daytime warming and the slope of nighttime warming, and the results are shown in Figure 1. The results of the analysis of covariance showed a significance result of 0.0517 for the slope of change in daytime warming and nighttime warming from 1982-2017, which did not pass the test of significance (P<0.05) and the test of extreme significance (P<0.01), but passed the test of 0.1. The basic principle of analysis of covariance is to combine linear regression with ANOVA by adjusting the experimental error terms of the group means and F-tests to test for significant differences between two or more adjusted means, and can be used to test for significant differences in the slope of diurnal warming change on long time series, as well as to visualise confidence intervals for the regression coefficients in a graph. According to the degree of significance of the slope of diurnal warming change, based on the opinions of other experts, this paper focuses on the influence of diurnal increase on the growth of different vegetation types. The asymmetry of diurnal warming is relatively weak in this study, so the asymmetry of diurnal warming is modified to be inconsistency. Based on the results of the analysis of covariance, we have censored the inconsistency in section 3.1.1 to make the study results more objective and accurate. Meanwhile, the main focus in the paper is to analyze the variability of daytime and nighttime warming on the growth of different types of vegetation and to reveal differences in the response mechanisms of vegetation in the transition zone areas to diurnal warming. Once again, we thank the reviewers for their valuable comments and hope that our revisions will meet your requirements.
Figure.1 Variation trend of Tmax and Tmin in Qinba mountains from 1982 to 2017
Point 2: The authors have added sections 2.3.1 Maximum value composite method and 2.3.2 ANUSPLIN interpolation method, which provide useful methodological information. However, the section on ANUSPLIN is far from providing the details required for scientific reproducibility. Also, in both these new sections, prior to each equation, the authors use the wording "The correlation coefficients are calculated as follows:", despite none of the equations being even remotely related to correlation coefficients. Such careless revisions call into question the rigor put into the rest of the manuscript.
Response 2:Thank you very much for your valuable suggestions. Parts 2.3.1 and 2.3.2 were added in the text based on comments from other reviewers. Due to an oversight on my part, parts 2.3.1 and 2.3.2 were not well presented in the text, causing ambiguity to reviewers and readers about the article, and the relevant coefficients were revised to be calculated equations. In addition, the ANUSPLIN interpolation method, as a common interpolation model, has been widely used for applications in climate data and can well improve the interpolation accuracy of meteorological data. The focus of this paper is to examine the effects of diurnal warming on vegetation growth and to explore the effects of daytime and nighttime warming on the growth of various vegetation types. Once again, we would like to thank the reviewers for their practical and effective suggestions on this paper and hope that our revisions will meet your requirements.

Reviewer 3 Report (Previous Reviewer 3)
Please see the attached.

Author Response
Thank you for giving me the opportunity to submit a revised draft of my manuscript titled ‘Climate change and diurnal warming: Impacts on the growth of different vegetation types in the North-South Transition Zone of China’ (Manuscript ID: Land-2034471) to Land. We appreciate the time and effort that you and the reviewers dedicated to providing you valuable feedback on my manuscript. We are grateful to the reviewers for the insightful comments on my paper. All these comments were valuable and extremely helpful for revising and improving our paper. We have studied the comments carefully and have made corrections, which we hope will meet with your approval. We have highlighted the changes within the manuscript.
Here is a point-by-point response to the reviewers’ comments and concerns.
Point 1: This manuscript has been revised carefully in the response to the previous comments. However,as the discussion section is still enumerating fragmentary outcomes from other studies, the discrimination of this study is not so clear. More intensive discussion on the finding from this research is required, in order to clarify the necessity and importance of this study.
Response 1: Thank you for your comments. We have made detailed changes and improvements to the discussion section. Once again, thank you for your practical and effective suggestions that have improved the quality of the paper overall. We hope that our modifications will meet your requirements.
Point 2:The captions of figures and tables should provide more detailed explanations.
Response 2: Thank you very much for your valuable comments. We have provided more detailed descriptions and explanations of the diagrams and tables in the text based on your suggestions.
Point 3: English through the whole text should be revised by a native speaker.
Response 3: Thank you very much for your valuable comments. We have asked native English-speaking experts to improve the overall language quality of the article based on your suggestions.

Reviewer 4 Report (Previous Reviewer 4)
Well done undertaking revisions so quickly and well. This manuscript looks much improved and I am recommending publication
Best wishes
Author Response
We would like to thank the editorial staff and the reviewers for their time in publishing this article, which has been revised and improved in detail based on the reviewers' comments. Once again, thank you for your hard work.

This manuscript is a resubmission of an earlier submission. The following is a list of the peer review reports and author responses from that submission.
Round 1
Reviewer 1 Report
The manuscript is interesting, but has a number of major issues that prevent publication in its present form. Some of these issues are:
1) The authors conclude that there is an asymmetry in diurnal temperature increase based on the fact that Tmax increased 0.6℃ per decade and Tmin increased 0.5℃ per decade (based on linear regression). However, these two values are relatively similar and each one has a confidence interval. The authors need to formally reject the null hypothesis that the rate of increase in Tmax and Tmin are equal before coming to the conclusion of asymmetrical warming.
2) The authors make mention of ANUSPLIN, but fail to provide any reference to this method or the inputs used. It is also unclear whether ANUSPLIN was applied to the daily data or the annual data.
3) The Mann-Kendall (M-K) test is mentioned on several occasions (including the abstract) and it even has its own section (2.3.1), but none of the results in the manuscript actually make use of the M-K test.
4) The authors state that "semi-monthly NDVI data were synthesized into monthly NDVI using the maximum synthesis method [46]". However, the paper referred to makes no mention of this "maximum synthetis method".
5) The reference numbering seems incorrect, e.g. [48-50] in the text actually refers to references [47-49].
6) The authors mention that they only consider the areas where the vegetation types have not changed, with is very important in the context of afforestation projects in the study region/period. However, from the results (Figures 5-7) it is unclear which regions (if any) were excluded due to vegetation type change.
Author Response
Reviewer 1:
Journal name: Land
Manuscript title: How does Diurnal Warming Affects the Growth of Different Vegetation Types in the North-South Transition Zone of China? An Empirical Study from the Qinlin-Daba Mountains
Manuscript ID: Land-1952477
Dear Editors and Reviewers:
Thank you for giving me the opportunity to submit a revised draft of my manuscript titled ‘How does Diurnal Warming Affects the Growth of Different Vegetation Types in the North-South Transition Zone of China? An Empirical Study from the Qinlin-Daba Mountains’ (Manuscript ID: Land-1952477) to Land. We appreciate the time and effort that you and the reviewers dedicated to providing you valuable feedback on my manuscript. We are grateful to the reviewers for the insightful comments on my paper. All these comments were valuable and extremely helpful for revising and improving our paper. We have studied the comments carefully and have made corrections, which we hope will meet with your approval. We have highlighted the changes within the manuscript.
Here is a point-by-point response to the reviewers’ comments and concerns.
The manuscript is interesting, but has a number of major issues that prevent publication in its present form. Some of these issues are:
Point 1:The authors conclude that there is an asymmetry in diurnal temperature increase based on the fact that Tmax increased 0.6℃ per decade and Tmin increased 0.5℃ per decade (based on linear regression). However, these two values are relatively similar and each one has a confidence interval. The authors need to formally reject the null hypothesis that the rate of increase in Tmax and Tmin are equal before coming to the conclusion of asymmetrical warming.
Response 1:Thank you very much to the editor for correcting this issue. Combined with the opinions of other review experts, we have modified the diurnal asymmetric warming to diurnal warming. This paper focuses on the effects of diurnal warming on vegetation growth under the background of global change, and reveals the differences in the responses of various vegetation types at site and raster scales to diurnal warming. At the same time, we carefully modified the heating rate of Tmax and Tmin in Section 3.1 to change the diurnal heating asymmetry into inconsistency, which is convenient for reviewers and readers to understand the full text, increase the readability of the article, and improve the reading quality and level of the article. Thank you again for your valuable advice. We hope you will agree to our amendment.
Point 2:The authors make mention of ANUSPLIN, but fail to provide any reference to this method or the inputs used. It is also unclear whether ANUSPLIN was applied to the daily data or the annual data.
Response 2:Thank you for your comments. We think your opinion is very reasonable. We have detailed the ANUSPLIN model in the Methods section. Also, references are cited to further increase the rigor of ANUSPLIN method citation and enhance the readability of the article. In addition, the ANUSPLIN interpolation method interpolates the data of rainfall, daily maximum temperature and daily minimum temperature at the annual scale in the Qinba Mountain with a resolution of 8km×8km, which we have explained in the article. Again, we would like to thank the reviewers for correcting the problems with the article.
Point 3: The Mann-Kendall (M-K) test is mentioned on several occasions (including the abstract) and it even has its own section (2.3.1), but none of the results in the manuscript actually make use of the M-K test.
Response 3:We are very grateful to the reviewers for their valuable comments. We have removed the Mann-Kendall (M-K) test and checked and removed all Mann-Kendall (M-K) from the paper. Once again, thank you for your correction on this issue. We hope you will agree to our amendment.
Point 4:The authors state that "semi-monthly NDVI data were synthesized into monthly NDVI using the maximum synthesis method [46]". However, the paper referred to makes no mention of this "maximum synthetis method".
Response 4: Thank you very much for your correction on this issue. The maximum synthesis method is not presented in detail in this paper because it is applied in the preprocessing process of vegetation NDVI data and is not the main method for analyzing diurnal warming on the growth of different vegetation types. We introduce the maximum value synthesis method section, while citing reference for supplementation, in order to further increase the readability and enhance the rigor and standardization of the article. We hope you will agree to our amendment.
Point 5:The reference numbering seems incorrect, e.g. [48-50] in the text actually refers to references [47-49].
Response 5:Thank you for your comments. In conjunction with your previous comments, we have carefully checked and examined the references in the paper to ensure that the citations are standardized. Once again, thank you for your correction on this issue.
Point 6:The authors mention that they only consider the areas where the vegetation types have not changed, with is very important in the context of afforestation projects in the study region/period. However, from the results (Figures 5-7) it is unclear which regions (if any) were excluded due to vegetation type change.
Response 6: Thank you very much for your valuable comments. In this study, in order to analyze the effect of diurnal warming on the growth of different vegetation types using multi-period vegetation type data, we obtained two vegetation type maps for the period 1982-2015 and processed them through ArcGIS 10.3 to acquire vegetation types with constant area. Reforestation has an important influence on the change of vegetation type area in the region, but the large-scale reforestation areas in China are mainly distributed in the Loess Plateau, the construction of the Three Northern Protection Forest System and other areas with more fragile ecological environment. The Qinba mountains are located in the transition area from the northern subtropical zone to the southern warm temperate zone, with better water and heat conditions, more abundant vegetation growth, and relatively small changes in vegetation in the region from afforestation activities. Furthermore, we used two vegetation types for screening to obtain invariant vegetation types, which can effectively circumvent the influence of silvicultural impacts on the study results. We obtained the 8km×8km raster data by resampling the vegetation type data, NDVI data, and climate data. The proportion of the area of vegetation types that did not change was large, which may have masked the changing vegetation types during the mapping process, and the resolution of the images may also have some influence. We apologize for any ambiguity this may have caused to reviewers and readers. Lastly, we counted areas where the vegetation type did not change when we did the statistics and analysis for each raster. At the same time, we also add related literature for supplementary explanation. Figures 5 to 7 are mainly visual representations of the second-order partial correlation of Tmax and Tmin with different types of vegetation NDVI in the Qinba Mountains from 1982-2015, reflecting spatially the intensity of the effect of diurnal warming on vegetation growth, therefore, the area of vegetation type change has less influence on the visualization of the study results. Once again, thank you for your valuable comments, your suggestions have an important role in improving the quality of the article. We hope that our modifications will meet your requirements.

Reviewer 2 Report
Dear editor, dear authors,
The purpose of the article is to show how global warming affects the vegetation according to the species present in the study area, which is a transition zone with a strong contrast in relief between the western and eastern parts.
The manuscript is interesting and very useful, but it suffers from the lack of clarity of certain definitions as well as certain methodological approaches. This is why I propose a major revision of the paper before publication.
Major revision:
Lines 263, 264 …between NDVI and Tmax, and NDVI and Tmin in Qinba…
The original version …. partial correlation results between NDVI and Tmax and Tmin in Qinba mountains vegetation…. is unclear. This confusion is systematic and must be corrected throughout the paper.
Clarify what is diurnal asymmetric warming
Lines 118, 119 not clear: 1) How diurnal asymmetric warming affects the growth of vegetation? 2) How does diurnal asymmetric warming affects typical vegetation type growth?
Lines 224-226, the formulation lacks rigor and should be homogeneous: rxy-1-2 is the bias correlation coefficient of x and y excluding variables 1 and 2; rx2-1 and ry2-1 are the second-order partial correlation coefficients of x, 2 and y, 2, excluding 1.
Line 229: r is the partial correlation coefficient: Which one?
How is characterized diurnal warming? From the mean diurnal temperature?
Paragraph 3.2.2. supposes that the definition (significance) of NDVI related to particular species is good enough to allow the correlation analysis of different vegetation types on diurnal warming.
Minor revision:
Lines 22, 62 NDVI not yet defined.
Author Response
Reviewer 2:
Journal name: Land
Manuscript title: How does Diurnal Warming Affects the Growth of Different Vegetation Types in the North-South Transition Zone of China? An Empirical Study from the Qinlin-Daba Mountains
Manuscript ID: Land-1952477
Dear Editors and Reviewers:
Thank you for giving me the opportunity to submit a revised draft of my manuscript titled ‘How does Diurnal Warming Affects the Growth of Different Vegetation Types in the North-South Transition Zone of China? An Empirical Study from the Qinlin-Daba Mountains’ (Manuscript ID: Land-1952477) to Land. We appreciate the time and effort that you and the reviewers dedicated to providing you valuable feedback on my manuscript. We are grateful to the reviewers for the insightful comments on my paper. All these comments were valuable and extremely helpful for revising and improving our paper. We have studied the comments carefully and have made corrections, which we hope will meet with your approval. We have highlighted the changes within the manuscript.
Here is a point-by-point response to the reviewers’ comments and concerns.
Point 1: Lines 263, 264 …between NDVI and Tmax, and NDVI and Tmin in Qinba…
Response 1: Thank you for your comments. We have revised and corrected Lines 263, 264. The ‘spatial analysis of second-order partial correlation results between NDVI and Tmax and Tmin in Qinba mountains vegetation from 1982 to 2015’ modify to ‘second-order partial correlation analysis of NDVI with Tmax and Tmin in the Qinba mountains from 1982 to 2015’. Also, we have made changes to the text in the figures to enhance the rigour of the article and to enable reviewers and readers to better understand the meaning of the text. Thank you once again for your comments. We hope you will agree to our amendment.
Point 2: The original version …. partial correlation results between NDVI and Tmax and Tmin in Qinba mountains vegetation…. is unclear. This confusion is systematic and must be corrected throughout the paper.
Response 2: Thank you very much for your suggestion. We have revised and refined the findings of the article to objectively and accurately describe the results of the second-order bias correlation between vegetation NDVI and Tmax and Tmin. Once again, thank you for your very valuable suggestions, which will further enhance the quality of the article and improve its scientific and rigorous nature. We hope you will agree to our amendment.
Point 3: Clarify what is diurnal asymmetric warming
Response 3: Thank you for your comments. We have revised the article in detail in response to your comments. Diurnal asymmetric warming refers to differences in the rate and magnitude of diurnal warming, which exhibit asynchrony. We have revised the article from diurnal asymmetric warming to diurnal warming, based on the comments of other reviewers. Thank you once again for your valuable comments.
Point 4: Lines 118, 119 not clear: 1) How diurnal asymmetric warming affects the growth of vegetation? 2) How does diurnal asymmetric warming affects typical vegetation type growth?
Response 4: Thank you very much for correcting this issue. We have revised and refined Lines 118,119 by modifying question 1 to how does diurnal warming affect vegetation growth at site and raster scales, promoting or suppressing it? Question 2 was modified to how diurnal warming affects the growth of typical vegetation types (Grass, Scrub, Deciduous broad-leaf forest, Evergreen deciduous broadleaf mixed forest, Evergreen broad-leaved forest, Coniferous forest, Crops), and whether there are differences in the response of different vegetation types to diurnal warming. In this section, the main focus is on two parts: firstly, the analysis of the effects of diurnal warming on vegetation growth based on site and raster scales, and secondly, the analysis of the effects of diurnal warming on the growth of different types of vegetation and the clarification of the differences in the response mechanisms of typical vegetation in the transition zone to diurnal warming. Based on these two parts, we have condensed the key scientific questions of this paper. Once again, thank you for your comments. We hope that our changes will meet your requirements.
Point 5: Lines 224-226, the formulation lacks rigor and should be homogeneous: rxy-1-2 is the bias correlation coefficient of x and y excluding variables 1 and 2; rx2-1 and ry2-1 are the second-order partial correlation coefficients of x, 2 and y, 2, excluding 1.
Response 5: Thank you for your comments. We have carefully revised and added to the formulas in Lines 224-226 to increase the rigour of the formulas and text. Once again, thank you for your suggestions as a reviewer and we hope that our revisions will meet your requirements.
Point 6: Line 229: r is the partial correlation coefficient: Which one?
Response 6: Thank you very much for your valuable advice from the reviewer. We have modified and improved Line 229 to further clarify the meaning of R. The meaning of R in this paper is the second-order partial correlation coefficient between Tmax and vegetation NDVI, and between Tmin and vegetation NDVI, respectively, which has been explained in this paper.
Point 7: How is characterized diurnal warming? From the mean diurnal temperature?
Response 7: Thank you for your comments. The diurnal warming data used in this paper are derived from the daily maximum and minimum temperatures on a daily basis. The daily maximum and minimum temperatures are extracted to form the average maximum and minimum temperatures on a monthly and annual scale, and the magnitude of diurnal warming is discerned through trend analysis. We used data on diurnal warming to carry out a second-order biased correlation analysis of the growth of different vegetation types in the Qinba Mountains, the data sources and processing of which are detailed in the article. Once again, thank you for your suggestions. We hope you will agree to our amendments.
Point 8: Paragraph 3.2.2. supposes that the definition (significance) of NDVI related to particular species is good enough to allow the correlation analysis of different vegetation types on diurnal warming.
Response 8: Thank you for your comments. The main objective of the study is to analyze the impact of diurnal warming on the growth of different vegetation types in the Qinba Mountains as a whole by extracting climatic data on the area of different vegetation types and diurnal warming at the site scale in the paragraph 3.2.2. The results of the study can be used as a reference for regional vegetation ecology research in the context of global change. Once again, thank you for your valuable comments. We hope that our modifications will meet your requirements.
Point 9: Lines 22, 62 NDVI not yet defined.
Response 9: Thank you for your comments. We have defined the Line 22 NDVI and written in detail about the NDVI at the front of the summary. Once again, thank you for your valuable comments.

Reviewer 3 Report
Please see the attached.

Author Response
Reviewer 3:
Journal name: Land
Manuscript title: How does Diurnal Warming Affects the Growth of Different Vegetation Types in the North-South Transition Zone of China? An Empirical Study from the Qinlin-Daba Mountains
Manuscript ID: Land-1952477
Dear Editors and Reviewers:
Thank you for giving me the opportunity to submit a revised draft of my manuscript titled ‘How does Diurnal Warming Affects the Growth of Different Vegetation Types in the North-South Transition Zone of China? An Empirical Study from the Qinlin-Daba Mountains’ (Manuscript ID: Land-1952477) to Land. We appreciate the time and effort that you and the reviewers dedicated to providing you valuable feedback on my manuscript. We are grateful to the reviewers for the insightful comments on my paper. All these comments were valuable and extremely helpful for revising and improving our paper. We have studied the comments carefully and have made corrections, which we hope will meet with your approval. We have highlighted the changes within the manuscript.
Here is a point-by-point response to the reviewers’ comments and concerns.
Point 1: It provides a significant literature review on the key concepts of this study, and the clear results and finding from the results. However; the discussion section is mainly focusing on the summary of this study and many deductions were explained subjectively without reference. So it is needed to get the overall discussion section more sophisticated.
Response 1: Thank you very much for your valuable comments. We have revised and improved the discussion section of the article based on your suggestions, and have also introduced references in the discussion section for additional clarification to improve the quality of the discussion as a whole and to increase the scientific quality of the article. Once again, thank you for your valuable suggestions. We hope that our revisions will meet your requirements.
Point 2: Explanation of the equations used in this study should be re-checked.
Response 2: Thank you very much for pointing this out. We have checked and corrected all the explanations of the equations in the article to ensure that each one is accurate and reasonable.
Point 3: The captions of figures and tables should provide more detailed explanations.
Response 3: Thank you very much for your suggestions. We have explained the figures and tables in the article in more detail to make the headings of the figures and tables more scientific and standardised and to increase the understanding of the article by reviewers and readers. Once again, thank you for your valuable comments.
Point 4: English through the whole text should be revised by a native speaker. For example, the use of 'diurnal' for meaning 'daily' rather than 'daytime' can mislead authors.
Response 4: Thank you very much for pointing out this problem. We have systematically checked and corrected the terminology in the article to make the terminology appearing in the article more scientific and standardised to avoid ambiguity to the readers. Also, Diurnal in this paper means day and night, not just daytime. Once again, thank you for your valuable comments and we hope that our changes will meet your requirements.

Reviewer 4 Report
This is a v good paper, strongly based in the existing literature with a good methodology and tackling a significant issue. It makes a useful contribution and sets the scene for further research in this area.
The literature surveyed is extensive, relevant and handled well in establishing the basis and parameters for this research.
The research data used is well-chosen and your analysis is quite exhaustive (see suggestion on this, below). The statistical approach is well thought out, and the results presented logically and convincingly.
Your discussion is very thoughtful and your findings and insights into areas for further research are valuable and will add to the literature in this area.
Some comments and suggestions
The following are minor comments and suggestions only:
1. Title: I wonder if you are underselling the paper. Given you are presenting research on a very significant impact of climate change, perhaps have that in the title? Something like: 'Climate change and diurnal warming: Impacts on the growth of different vegetation types in the North-South Transition Zone of China'.
2. Introduction: The Introduction is quite long, and in some sections contains material that could be shortened or moved to other sections.
For example - Line 93 - you begin to speak about the study area, but quite abruptly. Further down (Line 104 ff) you begin describing the study area in further detail, but then more directly set out the location etc. (Line 132 ff); but this is to some extent repeated (and with a location map) further down (Line 131 ff).
Perhaps delete some of the overlapping description from the Introduction, or move essential elements down into the Data and Methods section?
Minor editorial suggestions
Line 1: Delete the 's' in 'Affects'
Lines 41 - 43: Sentence beginning 'Vegetation is a natural...' - a v long sentence with several sub-clauses - perhaps break it into two or three shorter sentences?
Line 41: Perhaps change 'circles' to 'cycles'
Line 95 Rossi et al is actually ref 36, not ref 37;. Also the two authors of this article are Ross, S. & Isabel, N.; see also below regarding this ref - at Line 545
Line 119: Delete the 's' in 'affects'
Line 131 Capitalise the 's' in 'Study'
Line 285: 'Tab.1' I think should be 'Table.1'
Line 420: Perhaps re-word this sentence from 'In the future research process, we should...' to something like 'In future research we could (or aim to?) further develop the following: '
Line 430: Perhaps re-word this sentence form 'Global warming is an essential trend for future climate change...' to 'Global warming is an essential element (or component?) of future climate change...'
Line 539: Zhao et al 2018b - can't see a Zhao et al 2018a? Perhaps it should be simply Zhao et al 2018?
Line 545: 'Rossi, S. IsabelN.' - I think you mean 'Rossi, S. & Isabel, N.'
Line 574: Insert a space between 'Kiss" and 'O.'
Line 581: Journal name should be 'Geographical Research'
Reference list formatting
Some references have a semi-colon (';') between author's names, some do not!
Final suggestion
Just a thought, but your paper makes several valuable points; I wonder if it would be worth (if you have not already) considering a more popular version of the research, for consumption by a wider more general audience???
Author Response
Reviewer 4:
Journal name: Land
Manuscript title: How does Diurnal Warming Affects the Growth of Different Vegetation Types in the North-South Transition Zone of China? An Empirical Study from the Qinlin-Daba Mountains
Manuscript ID: Land-1952477
Dear Editors and Reviewers:
Thank you for giving me the opportunity to submit a revised draft of my manuscript titled ‘How does Diurnal Warming Affects the Growth of Different Vegetation Types in the North-South Transition Zone of China? An Empirical Study from the Qinlin-Daba Mountains’ (Manuscript ID: Land-1952477) to Land. We appreciate the time and effort that you and the reviewers dedicated to providing you valuable feedback on my manuscript. We are grateful to the reviewers for the insightful comments on my paper. All these comments were valuable and extremely helpful for revising and improving our paper. We have studied the comments carefully and have made corrections, which we hope will meet with your approval. We have highlighted the changes within the manuscript.
Here is a point-by-point response to the reviewers’ comments and concerns.
Point 1:Title: I wonder if you are underselling the paper. Given you are presenting research on a very significant impact of climate change, perhaps have that in the title? Something like: 'Climate change and diurnal warming: Impacts on the growth of different vegetation types in the North-South Transition Zone of China'.
Response 1: Thank you for your comments. We have made careful changes and improvements in the title of the paper based on your comments. The title was amended to ‘Climate change and diurnal warming: Impacts on the growth of different vegetation types in the North-South Transition Zone of China.’ Thank you once again for your valuable comments.
Point 2: Introduction: The Introduction is quite long, and in some sections contains material that could be shortened or moved to other sections. For example - Line 93 - you begin to speak about the study area, but quite abruptly. Further down (Line 104 ff) you begin describing the study area in further detail, but then more directly set out the location etc. (Line 132 ff); but this is to some extent repeated (and with a location map) further down (Line 131 ff). Perhaps delete some of the overlapping description from the Introduction, or move essential elements down into the Data and Methods section?
Response 2: Thank you very much for your valuable comments. We have trimmed the introductory section to rationalise the context and make the text more prescriptive. Also, we moved the geographical description of the study area to the data and methods department, enhancing information such as detailed geographic information descriptions. We hope you will agree to our amendment.
Point 3: Line 1: Delete the 's' in 'Affects'
Response 3: Thank you for your comments. We have revised the title of the paper in accordance with your requirements. The article's title has been corrected to ‘Climate change and diurnal warming: Impacts on the growth of different vegetation types in the North-South Transition Zone of China’. Thank you once again for your valuable comments. We hope you will agree to our amendment.
Point 4: Lines 41 - 43: Sentence beginning 'Vegetation is a natural...' - a v long sentence with several sub-clauses - perhaps break it into two or three shorter sentences?
Response 4: Thank you for your comments. We have split and revised Lines 41-43 according to your suggestion to make the logic more accessible and the language more concise, and also to increase the understanding of the editor and the readers and enhance the readability of the article. Once again, we thank the reviewers for their valuable suggestions.
Point 5: Line 41: Perhaps change 'circles' to 'cycles'
Response 5: Thank you very much for your valuable comments. We have corrected 'circles' to 'cycles' in line with your comments and hope that our changes will meet your requirements.
Point 6: Line 95 Rossi et al is actually ref 36, not ref 37; Also the two authors of this article are Ross, S. & Isabel, N.; see also below regarding this ref - at Line 545.
Response 6: Thank you for your comments. We have revised Line 95 and also corrected the author of the reference in Line 545. Once again, we thank the reviewer for raising this issue and we have meticulously checked all references and annotations in the article in the light of your comments. We hope you will agree to our amendment.
Point 7: Line 119: Delete the 's' in 'affects'
Response 7: Thank you for your comments. We have deleted the s in effects in this paper.
Point 8: Line 131 Capitalise the 's' in 'Study'
Response 8: Thank you for your comments. We have rewritten Study and also carefully checked the case in the article.
Point 9: Line 285: 'Tab.1' I think should be 'Table.1'
Response 9: Thank you for your comments. In the text, Tab.1 is changed to Table 1.
Point 10: Line 420: Perhaps re-word this sentence from 'In the future research process, we should...' to something like 'In future research we could (or aim to?) further develop the following: '
Response 10: Thanks you for your comments. Line 420 was changed and amended in the paper, such as ‘In the future research process, we should ….’ changed to ‘In future research we could further develop the following:’. Once again, we feel that you have made very practical suggestions. We hope you will agree to our amendment.
Point 11: Line 430: Perhaps re-word this sentence form 'Global warming is an essential trend for future climate change...' to 'Global warming is an essential element (or component?) of future climate change...'
Response 11: Thanks you for your comments. We have changed and amended line 430. ‘Global warming is an essential trend for future climate change…’ changed to ‘Global warming is an essential element of future climate change.’ Thank you again for your advice.
Point 12: Line 539: Zhao et al 2018b - can't see a Zhao et al 2018a? Perhaps it should be simply Zhao et al 2018?
Response 12: Thank you very much for your suggestion, we have made a change to Line 539.
Point 13: Line 545: 'Rossi, S. IsabelN.' - I think you mean 'Rossi, S. & Isabel, N.'
Response 13: Thank you very much for your suggestion. We have already made changes to Line 545. Once again, thank you for your valuable suggestions.
Point 14: Line 574: Insert a space between 'Kiss" and 'O.'
Response 14: Thank you very much for your advice. We have inserted a space between “Kiss” and “O”.
Point 15: Line 581: Journal name should be 'Geographical Research'
Response 15: Thank you very much for your suggestion. We have made a correction to Line 58.
Point 15: Reference list formatting. Some references have a semi-colon (';') between author's names, some do not!
Response 15: Thank you very much for your suggestion. We have made detailed corrections to the references.
Point 16: Final suggestion. Just a thought, but your paper makes several valuable points; I wonder if it would be worth (if you have not already) considering a more popular version of the research, for consumption by a wider more general audience???
Response 16: Thank you very much for your constructive comments. The study of diurnal asymmetric row warming on different vegetation types in the context of global change is the main focus of this paper, and the priorities for future research on vegetation in the Qinba Mountains proposed in the discussion section are important elements of the group's subsequent research. Once again, thank you for your valuable comments. In our future research, we will further carefully select topics based on your suggestions, and measure the response mechanisms of transitional vegetation type growth to climate change in the Qinba Mountains in multiple aspects and models. We hope that our revisions will meet your requirements.
